# A Personalized Approach to Percutaneous Coronary Interventions in the Left Main Coronary Artery—Is the Female Gender Associated with Worse Outcomes?

**DOI:** 10.3390/jpm11060581

**Published:** 2021-06-20

**Authors:** Marta Kałużna-Oleksy, Wojciech Jan Skorupski, Marek Grygier, Aleksander Araszkiewicz, Włodzimierz Skorupski, Stefan Grajek, Przemysław Mitkowski, Małgorzata Pyda, Maciej Lesiak

**Affiliations:** 1st Department of Cardiology, Poznań University of Medical Sciences, 61-848 Poznań, Poland; marta.kaluzna@wp.pl (M.K.-O.); mgrygier@wp.pl (M.G.); aaraszkiewicz@interia.pl (A.A.); w.skorupski@wp.pl (W.S.); stefan.grajek@skpp.edu.pl (S.G.); przemyslaw.mitkowski@skpp.edu.pl (P.M.); malgorzata.pyda@gmail.com (M.P.); maciej.lesiak@skpp.edu.pl (M.L.)

**Keywords:** bifurcation, distal left main stenosis, sex, PCI

## Abstract

There is still controversy whether the female gender is associated with worse outcomes after the percutaneous coronary intervention within the left main (LM PCI). This study aimed to examine gender-based differences in real-life LM PCI patients and present a gender-personalized LM PCI approach. Consecutively, 613 patients underwent LM PCI in our department from January 2015 to June 2019. Five hundred and thirty-three patients, with at least a one-year follow-up, were included in the study. There were 130 (24.4%) women and 403 (75.6%) men. Compared with men, women were older (70.0 ± 9.4 vs. 67.7 ± 9.2; *p* = 0.006) and had higher diabetes, hypertension, and chronic kidney disease rates. Left ventricle ejection fraction was higher in women (53.5 ± 9.4 vs. 49.5 ± 11.2; *p* = 0.001). Euroscore II and SYNTAX scores did not differ between the genders. However, we observed a trend towards more frequent use of complex PCI techniques in women (26.2% vs. 19.4%; *p* = 0.098). The overall periprocedural complication rates (10.0% vs. 7.7%; *p* = 0.406) and the periprocedural myocardial infarction rates did not differ. Contrast-induced nephropathy was more frequent in women (6.9% vs. 3.0%; *p* = 0.044). Long-term all-cause mortality did not differ (20% vs. 22.5%; *p* = 0.069). Both genders presented similar rates of periprocedural complications, and no significant differences in long-term all-cause mortality were revealed. Our results suggest that the female gender in LM PCI is not a predictor of adverse outcomes. Further studies are required to determine the optimal revascularization strategy in women.

## 1. Introduction

Coronary artery disease (CAD) remains the leading cause of morbidity and mortality in both men and women in developed countries [1]. Earlier research papers from the balloon angioplasty era showed that CAD was associated with worse survival in women than men [2,3]. However, the invention of drug-eluting stents (DES) reduced this sex gap [4,5,6]. Percutaneous coronary interventions (PCI) developments such as proper patient selection, device technology, stenting techniques, and medical therapy have made PCI a safe and effective alternative to coronary artery bypass grafting (CABG) for left main (LM) coronary artery disease [7,8,9]. However, despite the confirmed safety and efficacy of LM PCI [8,10], females stand for only about a quarter of patients in present research studies. In consequence, data in women undergoing LM PCI are scarce [11,12,13,14,15,16].

There is still controversy regarding whether female sex is associated with worse outcomes after LM PCI, and only a few data regarding this impact are available. Some studies reported no significant differences between the two genders [12,13,15], while others continued to report women to be at higher risk of major adverse cardiac events or death [14]. A recent meta-analysis has stated that women who underwent PCI for unprotected LM were at higher risk of major adverse cardiac events (MACE) and myocardial infarction (MI) compared to men [17]. Gender-based studies in patients undergoing PCI and CABG described higher in-hospital mortality and an increased rate of adverse outcomes in women than in men [2,3,18]. However, this difference may result from the fact that women tend to present to the hospital later than men. In addition, less favorable angiographic characteristics and comorbidities occur more frequently in this group [19,20]. In the SYNTAX trial, women undergoing PCI had a higher adjusted four-year mortality risk than men, while outcomes in the CABG group were comparable between genders [21]. As a result, gender became the major determinant in the SYNTAX Score II model that assists in selecting the best revascularization method [22]. In a recently published analysis from the EXCEL trial, sex was not an independent predictor of adverse outcomes. However, a worse trend with a higher frequency of periprocedural complications and long-term risk of MI in women undergoing PCI was observed [11].

A better understanding of gender-specific outcomes may potentially lead to developing individual revascularization strategies for a constantly growing population of women with CAD. Our study aimed to examine gender-based differences in real-life patients after LM PCI.

## 2. Materials and Methods

### 2.1. Study Population

Six hundred thirteen consecutive patients, who underwent LM PCI in our department from January 2015 to June 2019, were included in the initial analysis. The presence of at least 50% diameter stenosis of LM with or without the involvement of ostial left anterior descending artery (LAD), ostial left circumflex coronary artery (LCx), or both was the inclusion criterium. In patients with intermediate lesions, intravascular ultrasound imaging (IVUS) was used to confirm the significance of the stenosis, with a cut-off value of LM minimal lumen area of 6.0 mm^2^. We excluded from the study terminal patients whose expected survival was less than one year (Appendix A). After the Heart Team decision, PCIs were performed by experienced invasive cardiologists at a high-volume referral center with the Cardiac Surgery Department on-site.

### 2.2. Clinical and Angiographic Data

The clinical and angiographic data, including short- and long-term outcomes, were analyzed. Baseline clinical data were collected for each patient at the index procedure. The main procedural data with all periprocedural and in-hospital complications were collected and analyzed. Chronic kidney disease (CKD) was defined as an estimated glomerular filtration rate (eGFR) below 60 mL/min for three months or more. eGFR was calculated with Cockcroft–Gault equation.

All bifurcation lesions were assessed according to the Medina classification [23]. Patients with LM equivalent disease, i.e., distal bifurcation Medina 0-1-1, who presented < 70% stenoses of the ostial LAD or LCx without any evidence of ischemia in its myocardial distribution, were not included in the study [24]. Patients were treated to achieve complete revascularization of all major vessels with significant lesions in multi-stage procedures.

Periprocedural MI (Type 4a) was diagnosed based on the European Society of Cardiology Fourth Universal Definition of MI [25]. Contrast-induced nephropathy was defined as a serum creatinine increase of more than 25% or ≥ 0.5mg/dL (44 μmol/L) within 48 h [26]. Glycoprotein IIb/IIIa inhibitors, IVUS, or optical coherence tomography (OCT) were used at the operator’s discretion. However, IVUS or OCT imaging was used in 155 (29.1%) patients, and imaging findings were not analyzed. The antiplatelet regimen consisted of low-dose aspirin (75 mg daily) and clopidogrel (75 mg daily) for a minimum of 6 months after PCI with the intention of 12 months of dual antiplatelet therapy.

### 2.3. Study Endpoints

The primary short-term outcome of the study was the composite of in-hospital death or MI. At the same time, the long-term study endpoint was an all-cause mortality rate. The data were collected by telephone or based on the official records of the National Health Fund. The study conformed to the ethical guidelines of the 1975 Declaration of Helsinki and was granted ethics approval by the Institutional Review Board and the Bioethics Committee of the University.

### 2.4. Statistical Analysis

Statistical analysis was performed using STATISTICA 12 (Tibco Software Inc., Palo Alto, CA, USA). A standard descriptive statistic was applied in the analysis. All continuous variables are presented as means (standard deviation) or medians (interquartile range). The normality distribution was analyzed using the Shapiro–Wilk test. The statistical significance of differences was tested with the t-Student test or nonparametric U Mann–Whitney test. Categorical variables were reported as counts or percentages and compared by tests for proportions. The Kaplan–Meier method was used to calculate the survival probability at follow-up. The survival curves were compared with a log-rank test. A two-sided *p* value of <0.05 was considered significant for all the tests.

## 3. Results

### 3.1. Baseline Study Population Characteristics

From January 2015 to June 2019, we included 533 patients with LM PCI with available one-year follow-up. Of those, 403 (75.6%) were men and 130 (24.4%) were women. Patients’ baseline characteristics are presented in Table 1. Comparing to men, women were older (F vs. M: 70.0 ± 9.4 vs. 67.7 ± 9.2 years; *p* = 0.006), more frequently presented with arterial hypertension (90.8% vs. 80.9%; *p* = 0.009), CKD (42.3% vs. 30.8%; *p* = 0.015) and diabetes mellitus (45.4% vs. 34.0%; *p* = 0.019). By contrast, women were less likely to have history of previous MI (37.7% vs. 52.4%; *p* = 0.004), prior PCI in LCx (9.2% vs. 16.4%; *p* = 0.045). Left ventricle ejection fraction was higher in women (53.5 ± 9.4 vs. 49.5 ± 11.2; *p* = 0.001). Euroscore II values did not differ between two genders.

Coronary artery disease characteristics are shown in Table 2. We observed no major differences between the two genders in CAD characteristics. However, women tended to have the less advanced atherosclerotic disease with the SYNTAX score of 23.5 ± 9.3 vs. 24.8 ± 10.2; *p* = 0.301. Trifurcation lesions (7.7% vs. 14.1%, *p* = 0.054) and LM plus three-vessel disease (8.3% vs. 13.4%; *p* = 0.047) were less frequent in women. In addition, chronic total occlusions of right coronary artery were less frequent in women (10.0% vs. 21.1%; *p* = 0.005). LM lesions characteristics were similar in both sexes with no major differences in Medina types.

### 3.2. Procedure Details

No significant differences in the frequency of selected stenting techniques were registered, although we observed a trend towards more frequent use of the complex/two-stent techniques in women (26.2% vs. 19.4%; *p* = 0.098) (Table 3). All LM lesions were stented with second-generation DES. The number of stents and total length of implanted stents did not differ significantly between the two genders. Radiation dose was higher in men (1215 ± 812 vs. 1497 ± 884; *p* < 0.001). All LM procedures were carried out without left ventricular assist devices.

### 3.3. Outcomes

Periprocedural clinical outcomes are summarized in Table 4. The procedural success rates were high (99.2% vs. 99.8%; *p* = 0.984) and overall periprocedural complications rates (10.0% vs. 7.7%; *p* = 0.406) were similar in both groups. Contrast-induced nephropathy was significantly more frequent in women (6.9% vs. 3.0%; *p* = 0.044). Periprocedural mortality and MI type 4a rates did not differ between the groups.

The median follow-up was 1054 days (interquartile range: 662 days). Long-term all-cause mortality did not differ (20% vs. 22.5%; *p* = 0.069) (Figure 1).

## 4. Discussion

Even one in four or five patients with LM disease is a female [11,17]. In recent years, differences between men and women in LM PCI are an area of research. Previous literature data showed a worse prognosis in women after coronary revascularization with a higher risk of death and MI. It was attributed to older age, higher incidence of comorbidities (i.e., diabetes or arterial hypertension), and a higher risk profile of CAD [27,28]. However, recently published studies have suggested that gender does not significantly affect long-term patients’ prognosis [11,12,13,15,16] and that it was the higher incidence of comorbidities that influenced the outcomes. In that real-life registry, women, as compared to men, had different clinical and lesion characteristics. Women treated for LM disease had a higher number of comorbidities than men. Similar to analysis from the Excel study, women were older, had higher rates of diabetes, arterial hypertension, and CKD [11]. However, the frequency of these comorbidities in our real-life study was much higher than in the EXCEL subgroup analysis. The diabetes, hypertension, and CKD rates were 45.4% vs. 32.7%, 90.8% vs. 81.0%, and 42.3% vs. 28.1%, respectively [11]. Similar results, with higher rates of selected comorbidities in females than in men, were described in the MITO study (diabetes: 48.1% vs. 36.4%, CKD: 58% vs. 41.6%) [16] and the registry by Shin E-S et al. (hypertension: 69% vs. 60.2%) [15].

An analysis from the Excel trial showed that women had lower coronary atherosclerosis advancement [11]. Similar to the MITO registry, no major differences in the advancement of atherosclerotic disease (SYNTAX Score) were described in our study. The lower rate of LM plus three-vessel disease in women (8.3% vs. 13.4%; *p* = 0.047) was consistent with results shown in the registry by Shin E-S et al. (25.1% vs. 29.7%) and Excel trial analysis (10.2% vs. 18.9%) [11,15]. In the registry by Shin E-S et al., bifurcations occurred more often in men. In our study, such difference was not observed; however, the tendency to higher trifurcation occurrence was described [15]. True bifurcation lesions accounted for 38% of all bifurcations in our study.

Interestingly, despite no difference in the SYNTAX score, complex procedures were more common in women. Two-stent techniques were used more often, and the mean number of used stents was higher; however, these differences were not significant. The SYNTAX score does not also account for all anatomic complexities. This is consistent with MITO registry results where LM lesions in women characterized greater calcifications and needed rotational atherectomy more often despite no differences in the SYNTAX scores [16].

The successful use of radial access (associated with better outcomes) among women remained lower than in men [29]. No differences in the access site were described in our study, with 57.7% of radial access in women and 56.8% in men (*p* = 0.862). No differences in in-hospital bleeding rates were revealed. Wang et al. showed that radial access was used in 64% of women compared to 79% in men (*p* = 0.039). Authors claimed that the thinner radial arteries in women posed difficulties during PCI as well as postoperative care. Therefore, authors did not recommend using that access. However, one must remember that the study was conducted in the Chinese population.

We observed no significant difference in in-hospital mortality rates between the two genders. The periprocedural MI showed the tendency to appear more often in women (6.9% vs. 3.5%; *p* = 0.092). Similar results were described in the Excel trial, although with a significant difference (7.5% vs. 2.4%; *p* < 0.001) [11]. The results obtained in our study and the Excel trial differ from the numbers showed by Trabattoni et al., where the periprocedural MI rates in women and men were particularly high (12.3% vs. 14.8%; *p* = 0.480).

Contrast-induced nephropathy was significantly more often in females (6.9% vs. 3.0%; *p* = 0.044); however, this may result from a higher CKD rate at baseline rather than from the female sex itself [30]. Other gender-based LM studies did not report the frequency of contrast-induced nephropathy.

We also revealed no difference in long-term all-cause mortality rates (20% vs. 22.5%; *p* = 0.069). This observation was similar to the majority of previous LM PCI gender-based trials [11,12,13,15,16]. A worse prognosis in terms of mortality in women following LM PCI in the paper by Trabattoni et al. probably resulted from the large mean age difference between women and men (69.7 vs. 65.4 years; *p* < 0.001)—greater than in the others and our study [14].

The presented study is an analysis of a real-world cohort of patients and has some limitations. The first limitation involves the lack of a surgical group. However, the comparison of such a group with the CABG group was beyond the scope of this study. Secondly, although the presented study was a prospective registry, not all clinical data were available. Thirdly, this is a real-life study, and LM disease is not a homogenous disease. Outcomes are influenced by the location of the disease (ostial/shaft/bifurcation), the complexity of the lesion, and distal CAD. Finally, the presented study analyzed in-hospital, as well as the long-term follow-up; however, we were able only to show all-cause mortality rates.

## 5. Conclusions

In our real-life cohort of patients, comorbidities were more frequent in women. We observed no significant difference in short-term results and long-term all-cause mortality between the two genders. Our results suggest that the female gender in LM PCI is not a predictor of adverse outcomes. Further studies are required to determine the optimal revascularization strategy in women.

## Figures and Tables

**Figure 1 jpm-11-00581-f001:**
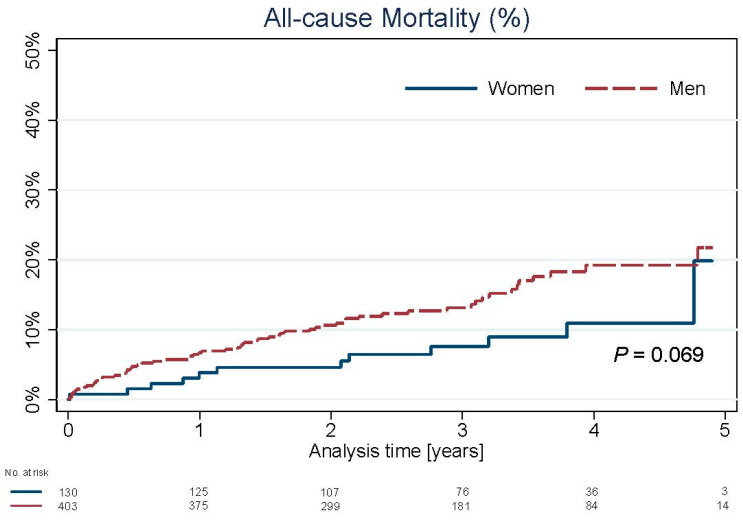
Kaplan–Meier analysis of all-cause mortality: women vs. men.

**Table 1 jpm-11-00581-t001:** Baseline characteristics by groups.

Variable	Women *n* = 130	Men *n* = 403	*p*-Value
Age (y)	70.0 ± 9.4	67.7 ± 9.2	0.006
BMI (kg/m^2^)	29.1 ± 4.9	28.2 ± 4.4	0.052
Hypertension	118 (90.8%)	326 (80.9%)	0.009
Hyperlipidemia	63 (48.5%)	202 (50.1%)	0.742
Chronic kidney disease	55 (42.3%)	124 (30.8%)	0.015
Diabetes	59 (45.4%)	137 (34.0%)	0.019
Stroke/TIA	7 (5.4%)	37 (9.2%)	0.171
COPD	8 (6.2%)	35 (8.7%)	0.357
Peripheral artery disease	19 (14.6%)	60 (14.9%)	0.939
Atrial fibrillation	16 (12.3%)	51 (12.7%)	0.917
Cigarette smoking (current)	46 (35.4%)	159 (39.5%)	0.407
Prior MI	49 (37.7%)	211 (52.4%)	0.004
Prior PCI LAD	24 (18.5%)	98 (24.3%)	0.167
Prior PCI LCx	12 (9.2%)	66 (16.4%)	0.045
Prior PCI RCA	33 (25.4%)	130 (32.3%)	0.139
Prior CABG	23 (17.7%)	92 (22.8%)	0.216
Clinical presentation:			
Stable angina	78 (60.0%)	240 (59.6%)	0.928
Unstable angina	35 (26.9%)	107 (26.6%)	0.933
NSTEMI	16 (12.3%)	38 (9.4%)	0.344
STEMI	2 (1.5%)	13 (3.2%)	0.312
LVEDD (mm)	47.1 ± 5.9	53.1 ± 7.3	<0.001
LVEF (%)	53.5 ± 9.4	49.5 ± 11.2	0.001
EuroScore II	2.32 ± 1.93	2.29 ± 1.87	0.826
Syntax Score:	23.5 ± 9.3	24.8 ± 10.2	0.301
0–22 (low)	63 (48.5%)	182 (45.2%)	0.511
23–32 (intermediate)	39 (30.0%)	124 (30.8%)	0.869
≥33 (high)	28 (21.5%)	94 (23.3%)	0.673

BMI—body mass index, TIA—transient ischemic attack, COPD—chronic obstructive pulmonary disease, MI—myocardial infarction, CAD—coronary artery disease, PCI—percutaneous coronary intervention, LAD—left anterior descending artery, LCx—left circumflex artery, RCA—right coronary artery, CABG—coronary artery bypass graft, LVEDD—left ventricular end diastolic diameter, LVEF—left ventricular ejection fraction.

**Table 2 jpm-11-00581-t002:** Coronary artery disease characteristics.

Variable	Women *n* = 130	Men *n* = 403	*p*-Value
LM distal	102 (78.5%)	323 (80.1%)	0.677
LM bifurcation	84 (64.6%)	256 (63.5%)	0.828
LM trifurcation	10 (7.7%)	57 (14.1%)	0.054
LM calcification	19 (14.6%)	55 (13.6%)	0.781
LAD disease (not ostial)	8 (6.2%)	34 (8.4%)	0.401
LCx disease (not ostial)	9 (6.9%)	22 (5.5%)	0.535
Protected LM	12 (9.2%)	65 (16.1%)	0.052
RCA recessive (a)	8 (6.2%)	30 (7.4%)	0.619
RCA with significant stenosis (b)	20 (15.4%)	60 (14.9%)	0.890
RCA total occlusion (c)	13 (10.0%)	85 (21.1%)	0.005
Lack of RCA support (a + b + c)	41 (31.5%)	175 (43.4%)	0.016
CTO of RCA with collateral circulation from LCA	9 (6.9%)	55 (13.6%)	0.040
Extent of diseased vessels:			
LM plus 2-vessel disease	28 (21.5%)	109 (27.0%)	0.211
LM plus 3-vessel disease	9 (8.3%)	54 (13.4%)	0.047
Medina classification:	*n* = 84	*n* = 256	
1-0-0	22 (26.2%)	84 (32.8%)	0.256
1-0-1	10 (11.9%)	35 (13.7%)	0.678
1-1-0	30 (35.7%)	76 (29.7%)	0.301
1-1-1	22 (26.2%)	61 (23.8%)	0.662

LM—left main, LAD—left anterior descending artery, LCx—left circumflex artery, RCA—right coronary artery, CTO—chronic total occlusion, LCA—left coronary artery.

**Table 3 jpm-11-00581-t003:** Left main PCI procedure characteristics.

Variable	Women *n* = 130	Men *n* = 403	*p*-Value
PCI success	129 (99.2%)	402 (99.8%)	0.984
Number of stents	1.82 ± 0.93	1.67 ± 0.82	0.097
Total length of implanted stents [mm]	39.7 ± 22.8	38.7 ± 22.5	0.612
Radiation time [min]	17.9 ± 8.9	17.8 ± 9.6	0.647
Radiation dose [mGy]	1215 ± 812	1497 ± 884	<0.001
Contrast volume [mL]	242.6 ± 99.9	254.9 ± 92.4	0.166
Arterial access site			
Radial	75 (57.7%)	229 (56.8%)	0.862
Femoral	55 (42.3%)	174 (43.2%)	
Stenting LM only	17 (13.1%)	47 (11.7%)	0.666
Stenting LM bifurcation			
One-stent technique	79 (60.8%)	278 (69.0%)	0.083
Two-stent technique	34 (26.2%)	78 (19.4%)	0.098
Two-stent techniques:	*n* = 34	*n* = 78	
Crush	16 (47.1%)	30 (38.5%)	0.395
DK-crush	2 (5.9%)	12 (15.4%)	0.162
Cullote	1 (2.9%)	0 (0.0%)	0.668
T-stenting	6 (17.6%)	16 (20.5%)	0.726
Provisional stenting	9 (26.5%)	20 (25.6%)	0.927

PCI—percutaneous coronary intervention, LM—left main, DK-crush—double kissing crush technique.

**Table 4 jpm-11-00581-t004:** Periprocedural outcomes.

Variable	Women *n* = 130	Men *n* = 403	*p*-Value
Myocardial infarction	9 (6.9%)	14 (3.5%)	0.092
In-hospital death	0 (0.0%)	2 (0.5%)	0.984
Stroke	0 (0.0%)	1 (0.2%)	0.551
Tamponade	0 (0.0%)	2 (0.5%)	0.984
Pulmonary edema	0 (0.0%)	1 (0.2%)	0.551
Dissection of aorta	0 (0.0%)	1 (0.2%)	0.551
Perforation of femoral artery	0 (0.0%)	2 (0.5%)	0.984
Contrast-induced nephropathy	9 (6.9%)	12 (3.0%)	0.044

## Data Availability

The data presented in this study are available on request from the corresponding author.

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
