# Peer review of "A Personalized Approach to Percutaneous Coronary Interventions in the Left Main Coronary Artery—Is the Female Gender Associated with Worse Outcomes?"

_jpm, 2021, doi:10.3390/jpm11060581_

Round 1
Reviewer 1 Report
In this retrospective study Kałużna-Oleksy et al, examined gender-based differences in real-life LM PCI patients and presented a gender-personalized LM PCI approach.
There are series of questions that to my opinion need some clarification.
1/ The purpose of the paper and the novelty of the study should be emphasized by the authors
2/ Although a retrospective analysis, it might have been necessary to obtain an ethical agrement to review all the patient's files.
3/ please provide a flowchart of eligible but not included patients excluded patients (patients that denied the use of their data file).
Author Response
Ms. Michelle Fang
Assistant Editor
JPM Editorial Office
10.06.2021
Dear Professor Fang,
Manuscript number: JPM-1247459
We sincerely thank the reviewers for valuable and insightful comments. These remarks have helped us to improve our work. Our responses to the reviewers comments are given in a point-by-point manner below. The changes in the manuscript are marked up using the “Track Changes” function.
We hope you find the revised version appropriate and worth publishing in Journal of Personalized Medicine.
Sincerely,
Wojciech Skorupski
I Department of Cardiology,
Poznań University of Medical Sciences, Poznań, Poland
Responses to reviewers’ comments:
We sincerely thank the Reviewers for valuable and insightful comments.
Reviewer #1:
Comment 1: The purpose of the paper and the novelty of the study should be emphasized by the authors.
Response: We appreciate the reviewer’s comment. There is still controversy regarding whether female sex is associated with worse outcomes after LM PCI, and only few data regarding this impact are available. A better understanding of sex-specific outcomes may potentially lead to the development of individual revascularization strategies for a constantly growing population of women with coronary artery disease. We aimed to examine gender-based differences in real-life cohort of patients after LM percutaneous coronary interventions.
The following sentences are included in the "Introduction" section:
“A better understanding of gender-specific outcomes may potentially lead to developing individual revascularization strategies for a constantly growing population of women with CAD. Our study aimed to examine gender-based differences in real-life patients after LM PCI.”
Comment 2: Although a retrospective analysis, it might have been necessary to obtain an ethical agreement to review all the patient's files.
Response: We sincerely thank the reviewer for this comment. Our study was granted the necessary ethical approval by the Institutional Review Board and the Bioethics Committee of the University. In our center, the care for patients undergoing LM PCI includes telephone contact with patients after about 6 and 12 months, which is a standard procedure. The approval of the Bioethics Committee covers all routine procedures.
Comment 3: Please provide a flowchart of eligible but not included patients excluded patients (patients that denied the use of their data file).
Response: Five hundred thirty-three patients, with at least 1-year follow-up, were included in the study. The presence of at least 50% diameter stenosis of LM with or without the involvement of ostial left anterior descending artery (LAD), ostial left circumflex coronary artery (LCx), or both was the inclusion criterium. We excluded from the study terminal patients whose expected survival was less than one year, i.e. patients with severe coexisting diseases with a very high risk of mortality, e.g. advanced oncology disease with end-stage renal disease. None of the patients forbade the use of their data files. As mentioned above, standard of care in our center includes telephone contact after LM PCI at about 6 and 12 months after procedure.
We have added the requested flowchart as supplementary material.

Reviewer 2 Report
Congratulations on a nicely performed study. Please perform a check of the paper by an English expert, as some of the sentences are not clear. For example the first sentence of the discussion.
Author Response
Ms. Michelle Fang
Assistant Editor
JPM Editorial Office
Dear Professor Fang,
Manuscript number: JPM-1247459
We sincerely thank the reviewers for valuable and insightful comments. These remarks have helped us to improve our work. Our responses to the reviewers comments are given in a point-by-point manner below. The changes in the manuscript are marked up using the “Track Changes” function.
We hope you find the revised version appropriate and worth publishing in Journal of Personalized Medicine.
Sincerely,
Wojciech Skorupski
I Department of Cardiology,
Poznań University of Medical Sciences, Poznań, Poland
Responses to reviewers’ comments:
We sincerely thank the Reviewers for valuable and insightful comments.
Reviewer #2:
Comment 1: Congratulations on a nicely performed study. Please perform a check of the paper by an English expert, as some of the sentences are not clear. For example the first sentence of the discussion.
Response: We appreciate the reviewer’s comments. We have corrected the first sentence of the discussion (below) and also other sentences which were to complex. The text has been reviewed by a native speaker.
“Even one in four or five patients with LM disease is a female.”

Reviewer 3 Report
The present manuscript provides analysis of consecutive 613 patients underwent LM PCI from January 2015 to June 2019 aiming to examine gender-based differences in real-life LM PCI patients and present a gender-personalized LM PCI approach. The author provides gender differences in patients with LM PCI suggesting that both genders presented similar rates of periprocedural complications, and no significant differences in long-term all-cause mortality
Comments:
- Please provide number of patients in both groups for the respective time points in the KM analysis.
- Conclusions: please use “one-year follow-up” instead long-term.
- Conclusions: that can be drawn from the present analysis are hypothetical, given the study design and statistical methods. The authors cannot conclude that the is a trend toward higher mortality in men.
Author Response
Ms. Michelle Fang
Assistant Editor
JPM Editorial Office
Dear Professor Fang,
Manuscript number: JPM-1247459
We sincerely thank the reviewers for valuable and insightful comments. These remarks have helped us to improve our work. Our responses to the reviewers comments are given in a point-by-point manner below. The changes in the manuscript are marked up using the “Track Changes” function.
We hope you find the revised version appropriate and worth publishing in Journal of Personalized Medicine.
Sincerely,
Wojciech Skorupski
I Department of Cardiology,
Poznań University of Medical Sciences, Poznań, Poland
Responses to reviewers’ comments:
We sincerely thank the Reviewers for valuable and insightful comments.
Reviewer #3:
Comment 1: Please provide number of patients in both groups for the respective time points in the KM analysis.
Response: We followed the recommendations. We have added the number of patients in both groups for the respective time points in the Kaplan-Meier analysis.
Comment 2: Conclusions: please use “one-year follow-up” instead long-term.
Response: In our study we have included 533 patients with at least 1-year follow-up. As presented in the Kaplan-Meier chart, most patients were followed up for more than one year. However, if the reviewer's wishes remain so, we would have to rearrange the calculations.
Comment 3: Conclusions: that can be drawn from the present analysis are hypothetical, given the study design and statistical methods. The authors cannot conclude that the is a trend toward higher mortality in men.
Response: We appreciate the reviewer’s comment. We have toned down the conclusions. The above-mentioned wording has been removed from the manuscript.

Round 2
Reviewer 1 Report
The authors met all the requirements. I recommend publishing the manuscript in this form.